# Effects of different types of sensory signals on reaching performance in persons with chronic schizophrenia

**Paul Sin-Bao Huang[1], Chiung-Ling Chen[2,3], Kwok-Tak Yeung[2,3], Ming-Yi Hsu[4,5], Sok-Wa Wan[6], Shu-Zon Lou[2,3]***

**1** Department of Palliative Care, Changhua Christian Hospital, Changhua, Taiwan, **2** Department of Occupational Therapy, Chung Shan Medical University, Taichung, Taiwan, **3** Occupational Therapy Room, Chung Shan Medical University Hospital, Taichung, Taiwan, **4** Department of Nursing, Chung Shan Medical University, Taichung, Taiwan, **5** Department of Nursing, Chung Shan Medical University Hospital, Taichung, Taiwan, **6** Long Ching Centre of Fuhong Society of Macau, Macau, China

* szlou@csmu.edu.tw

**Data Availability Statement:** All relevant data are within the manuscript and its Supporting Information files.

## Abstract

Previous studies have reported movement abnormalities in persons with schizophrenia. This study aimed to examine the differences between persons with chronic schizophrenia and healthy control participants in reaching movement and the effects of sensory signals on reaching performance in persons with chronic schizophrenia. A counter-balanced repeated-measures design was employed. Twenty persons with schizophrenia and 20 age- and gender-matched control participants were recruited in this study. Reaching performance was measured in three types of sensory signal conditions (visual, auditory, and no signal), i.e., two externally triggered and one self-initiated movement were assessed in reaction time/ inter-response interval, movement time, peak velocity, percentage of time in which peak velocity occurred, and movement units. The results revealed significant main effects of group in reaction time/inter-response interval (p = 0.003), movement time (p < 0.001), peak velocity (p < 0.001), and movement units (p < 0.001). The persons with chronic schizophrenia demonstrated slower response to signals and in self-initiated movement, increased movement time, and less forceful and less smooth movement compared to healthy control participants when performing the reaching task. The interaction effect between group and signal in reaction time/inter-response interval was also significant (p < 0.001). The inter-response interval for self-initiated reaching was the shortest in healthy controls. Conversely, the inter-response interval for self-initiated reaching was the longest in persons with schizophrenia. The main effect of the signal on movement time was significant (p < 0.001). The movement time of reaching was longer in response to the auditory signal than in response to visual or self-initiated. The differences in percentages of time in which peak velocity occurred between persons with schizophrenia and healthy controls (p > 0.01) and across the three conditions (p > 0.01) were non-significant. Neither duration of illness nor antipsychotic dosage was significantly associated with reaching performance (all p > 0.01). In conclusion, these findings indicate that reaching movement in persons with chronic schizophrenia is slower, less forceful, and less coordinated compared to healthy control

**Funding:** SBH and CLC received award to the study. The study was funded by research grants from Chung Shan Medical University (www.csmu.edu.tw) and Changhua Christian Hospital (www.cch.org.tw) (CSMU-CCH-105-05). The funders had no role in study design, data collection and analysis, decision to publish, or preparation of the manuscript.

**Competing interests:** The authors have declared that no competing interests exist.

**Abbreviations:** RT, reaction time; IRI, inter-response interval; MT, movement time; PV, peak velocity; PPV, percentage of time where peak velocity occurred; MU, movement unit; SMA, supplementary motor area.

participants. In addition, persons with chronic schizophrenia also had shorter inter-response interval for self-initiated movement and shorter movement time in auditory signal condition, independent of duration of illness and antipsychotic dosage.

## Introduction

Persons with chronic schizophrenia exhibit a range of social, cognitive, emotional, and movement dysfunctions [1, 2]. Two-thirds of chronic schizophrenia patients suffer from a neuroleptic-induced movement disorder [3]. However, movement disorders were also observed in the pre-neuroleptic era and in patients who were never exposed to antipsychotic medications [4]. The prevalence rate of at least one motor sign was 66% for the first episode, never-mediated patients [5]. An 8-year follow-up study of chronic schizophrenia inpatients found that the proportion of movement disorder-free population remained the same over 8 years, and doses of antipsychotic drugs had no effect on the severity of neuroleptic-induced movement disorders [6]. These results support that movement disorders in schizophrenia may be related to the pathophysiology of psychotic disorders and therefore cannot be attributed entirely to the adverse effects of neuroleptic medication.

The movement disorders include diverse motor abnormalities with varying degrees of severity. The common types of movements in persons with schizophrenia are stereotypic movements with no actual function, such as hand flapping, rocking, or pacing [7], and catatonic unresponsive movement [8]. Motor symptoms in schizophrenia are not restricted to hyperkinetic and hypokinetic movement disorders, i.e., abnormal involuntary movements and parkinsonism. Higher motor functions, such as coordination, motor planning, and sequencing of complex motor acts, may be impaired [9].

Previous studies have reported that persons with schizophrenia show slower motor performance in the pegboard tasks [10, 11], poorer performance in the finger tracking task [12], and impaired motor dexterity on a finger movement test [11] compared to normal controls. All those tasks measure the speed and accuracy of the hand movement performance. In addition, the gross movement performance of the arm, such as reaching, was also slower and less direct in persons with schizophrenia compared to those of normal controls [13]. Arm reaching is a frequently performed motor behavior and an important element in many activities of daily living. Therefore, improving upper extremity function through arm reaching training is a common goal of rehabilitation [14, 15].

Changing the object size and its distance is one method has been found to enhance reaching performance in persons with schizophrenia [13]. It is believed that the movement performance improves when an individual focuses his/her attention externally on the movement outcome rather than internally on the body movement itself [16]. Therefore, it can be assumed that an external reaching target with light or sound signals could improve reaching performance.

However, whether different types of sensory signals influence reaching performance in persons with schizophrenia is unclear. The purpose of this study was to examine the reaching movement in persons with chronic schizophrenia compared to healthy control participants and to examine the effects of sensory signals on reaching performance. In addition, the relationship between reaching performance and both duration of illness and antipsychotic dosage was also examined in persons with schizophrenia. We hypothesized that persons with schizophrenia would have slower, less forceful, and less coordinated movements compared to healthy

control participants. Second, we hypothesized that visual signal would induce faster and more coordinated movements in persons with schizophrenia.

## Methods

### Research design

A two-way mixed design with one independent variable (group) and one repeated variable (signal) was used to determine the effects of different types of signal guidance on reaching performance.

### Participants

This study utilized a convenience sample comprising 20 persons with chronic schizophrenia (10 men and 10 women) and 20 gender- and age-matched healthy controls. All persons with schizophrenia diagnosed according to the Diagnostic and Statistical Manual of Mental Disorders Fourth Edition (DSM-IV) were recruited from a day care center in a psychiatric hospital. They all received atypical antipsychotics and attended the rehabilitation program daily in the center. The antipsychotic doses were collected for each patient and converted to chlorpromazine equivalents (CPZE) [17]. The rehabilitation program included physical conditioning as well as activities of daily living, social skills, time management, and prevocational training. The healthy controls were recruited from hospital staff and family members in the rehabilitation unit of the hospital. The inclusion criteria for the participants were being between 20 and 65 years old and being able to communicate effectively. Their Mini-Mental State Examination scores were greater than or equal to 25. Participants with signs or symptoms of other mental diseases, neurological diseases, orthopedic symptoms, or a history of alcohol or drug abuse were excluded.

### Ethical statement

A senior occupational therapist of the day care center assessed the capacity of the patients to understand and provide informed consent. All participants were informed about the purpose and procedures of the study, and they gave written informed consent before the start of the study. The study was approved by Tsaotun Psychiatric Center Institutional Review Board (102001s). In addition, the participant who appeared in Fig 1 in this manuscript had given written informed consent (as outlined in PLOS consent form) to publish.

### Instruments and kinematic measures

A self-designed Interactive Sound and Light Eye-Hand Coordination Training System was used to provide signal guidance for arm reaching. This system's hardware includes one start button, seven cylindrical bottles, and a computer. The start button was used to record the initiation and termination of movement in the reaching task. Each cylindrical bottle contained one speaker to provide auditory (a single sound) signal and a light-emitting diode tube to provide a visual (light) signal as the reaching guide. An infrared sensor was mounted on the top of the bottle to detect the touch of the hand and the light or sound signal stopped by the touch. For no signal condition, a light or sound feedback was provided when the hand touched the top of the bottle. The Visual Basic computer program developed by the researcher was used to display signals and record the reaction and total movement time of the reaching task. The reaching task included arm reaching to touch the target bottle and arm returning to touch the start button.

The velocity profile of arm reaching was recorded at a sampling frequency of 50 Hz using an ultrasonic 3-D motion analysis system (Zebris, CMS-HS V10, Medizintechnik GmbH, Germany). One triangle marker attached to the dorsum of the hand, was used to measure the arm reaching velocity. The WinData software (Zebris, CMS-HS, Medizintechnik, GmbH, Germany) was used to analyze the reaching velocity profile.

Reaching performance was measured in terms of reaction time/inter-response interval (RT/IRI), movement time (MT), peak velocity (PV), percentage of time where peak velocity occurred (PPV), and movement units (MU). RT/IRI and MT of total movement (reaching out to touch and return) were measured by the Interactive Sound and Light Eye-Hand Coordination Training System, and the other velocity parameters (PV, PPV, and MU) of reaching out to touch were measured by Zebris. RT was the duration between the onset of a stimulus and the initiation of the arm movement (the release of the start button). RT is the ability to detect, process, and respond to a stimulus. A shorter RT means a higher speed of early neural processing of a stimulus. For no signal condition, the first RT was the duration between the

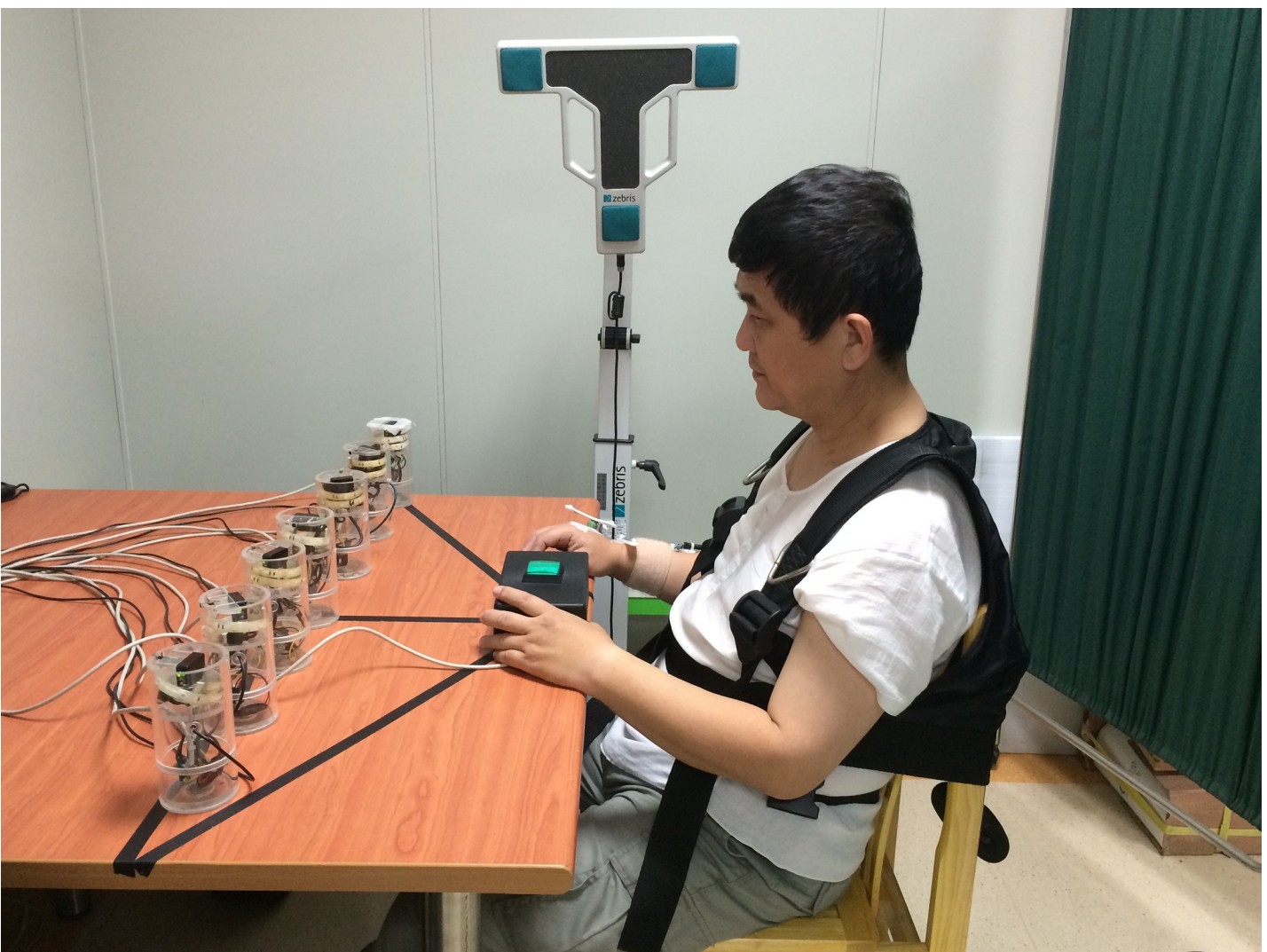

**Fig 1. Experimental set-up of the reaching task.** A restraint harness was worn to limit the trunk movement. The start button was located on the edge of the table in the sagittal midline. Cylindrical bottles that were the furthest were placed at 60∘from the sagittal midline on both sides of the table at a maximal arm length. The other five cylindrical bottles were placed in a straight line between the two furthest cylindrical bottles.

experimenter's starting command and the initiation of the arm movement, while the subsequent six IRIs represented the duration from the start button pushed (termination of the previous round of movement) to the start button released (initiation of the subsequent arm reaching). This movement was self-initiated. MT was the duration of total movement from the initiation of the arm movement to the termination of the movement (back to push the start button). A shorter MT indicated a faster movement. PV was the highest value of the velocity during the arm reaching out (hand touches the top of the bottle). A higher PV indicated greater force generation [18]. PPV was the percentage of time between the onset and the end of the arm reaching out when peak velocity occurs. The single peaked velocity profile of a typical arm reaching is a bell-shaped curve that includes one acceleration and one deceleration phase. The acceleration phase represents the portion of the movement under preplanned control, and the deceleration phase represents the feedback-controlled portion of the movement [19–21]. PPV is the proportion of movement time spent on the acceleration phase. A higher PPV means the arm reaching is more preplanned and more dependent on the feedforward control, while a lower PPV means the arm reaching is more dependent on sensory feedback [22, 23]. In this study, MU was defined by the velocity profile of movement, and it consisted of one acceleration phase and one deceleration phase. Fewer MUs mean a smoother movement, while more MUs mean less smooth movement [24]. For the velocity measurement, the onset of arm reaching out was defined as the time point at which the hand's velocity first reached 3% of its peak value. The end of arm reaching out was defined as the time point at which the hand's velocity decreased to 3% of its peak value when the hand touched the bottle.

## Procedures

Each participant sat with hip and knee joints flexed to 90˚ and feet supported in front of the experimental table. A restraint harness was worn to limit the trunk movement. The start button was located on the edge of the table in the sagittal midline. Cylindrical bottles that were the furthest were placed at 60◦from the sagittal midline on both sides of the table at maximal arm length (determined for each participant measured from the medial border of the axilla to distal wrist crease during arm reaching movement). The other five cylindrical bottles were placed in a straight line between the two furthest cylindrical bottles (Fig 1).

All participants were right-handed and used their right arm to execute the reaching task in this study. At the beginning of the reaching task, the participants' right upper arm was close to the body with elbow flexion of 90◦, forearm pronation, and wrist and fingers in neutral extension. Four fingers were closed together and placed on the start button. Upon receiving the visual/auditory signal or the experimenter's starting command, the participants moved their arms to reach the signal-emitting bottle. They touched the top sensor of the bottle as quickly and accurately as possible, immediately returning to the start button. The signals appeared randomly across the seven bottles. For no signal condition, the participants first moved their arms to touch the furthest bottle on the left in response to the experimenter's start command and then self-initiated the subsequent movements to touch the bottles in the order of the placement. Each participant performed three sessions (auditory, visual, and no signal) with 3 trials in each session and seven rounds of arm reaching in each trial. The order of the sessions was random and counter-balanced. The experimental task was clearly explained first, and participants could practice before each session until they felt familiar with it.

## Data analysis

We used a two-way mixed analysis of variance (ANOVA) with one between factor (Group: persons with schizophrenia and healthy control participants) and one within factor (Signal:

no, visual, and auditory signals) to test the differences between two groups and three types of sensory signals. Post-hoc comparisons with Bonferroni adjustments were then conducted. In addition, Pearson's correlations were conducted to examine the relationship between reaching performance and both duration of illness and antipsychotic dosage. The means of seven rounds of reaching movement for one trial and for three trials were calculated for further descriptive and inferential statistics. Statistical analyses were performed using the Statistical Package for the Social Sciences version 20.0 (SPSS, Inc., Chicago, IL, USA). To account for multiple comparisons, statistical significance was set at .01. For all dependent variables, when the sphericity assumption was violated the Greenhouse-Geisser correction was used to report F and p values.

## Results

Twenty persons with chronic schizophrenia and 20 healthy subjects participated in this study. Table 1 shows the demographic characteristics of the participants. The mean age was 39.63 years (SD = 10.12 years) for the patients and 39.09 years (SD = 9.47 years) for the control group.

The main effects of group on RT ($F_{(1, 38)}$ = 19.73, $p < 0.001$), MT ($F_{(1, 38)}$ = 17.58, $p < 0.001$), PV ($F_{(1, 38)}$ = 35.93, $p < 0.001$), and MU ($F_{(1, 38)}$ = 24.42, $p < 0.001$) were significant. The persons with chronic schizophrenia showed significantly greater RT, MT, and MU but lower PV compared to healthy subjects when performing the reaching task. An interaction effect between group and signal on RT was significant ($F_{(2, 76)}$ = 10.92, $p < 0.001$). The shortest IRI of reaching emerged for no signal condition in healthy controls. Conversely, the IRI of reaching was longest for no signal condition in persons with schizophrenia. The main effect of signal on MT was significant ($F_{(1.59, 60.50)}$ = 114.17, $p < 0.001$). The MT of reaching was significantly longer in response to the auditory signal compared to no or visual signal. The differences in PPV between persons with schizophrenia and healthy subjects ($F_{(1, 38)}$ = 0.45, $p > 0.01$) and among three types of signals ($F_{(2, 76)}$ = 3.07, $p > 0.01$) were non-significant (Fig 2). The results of Pearson's correlations showed that reaching performance neither significantly correlated with duration of illness nor with antipsychotic dosage (all p values > 0.01, the Pearson's correlation coefficient rs range from -0.418 to 0.443) (Table 2).

## Discussion

The findings of this study showed that persons with chronic schizophrenia had, as hypothesized, significantly slower RT/IRI and MT. Their reaching movement was also less forceful and smooth compared to healthy controls. This result is consistent with the previous studies

**Table 1. Demographic data of the participants.**

| Variable | Controls Mean (SD) | schizophrenia Mean (SD) |
|---|---|---|
| Age (years) | 39.09 (9.47) | 39.63 (10.12) |
| Gender, male/female | 10/10 | 10/10 |
| Education (years) | 15.10 (2.63) | 13.10 (2.36) |
| Onset age (years) | N/A | 21.75 (5.87) |
| Duration of illness (years) | N/A | 17.88 (7.24) |
| Dosage of antipsychotics (CPZE[a], mg) | N/A | 322.30 (195.33) |
| MMSE[b] | N/A | 28.60 (1.23) |

[a]CPZE: Chlorpromazine equivalent dose
[b]MMSE: Mini-Mental State Examination

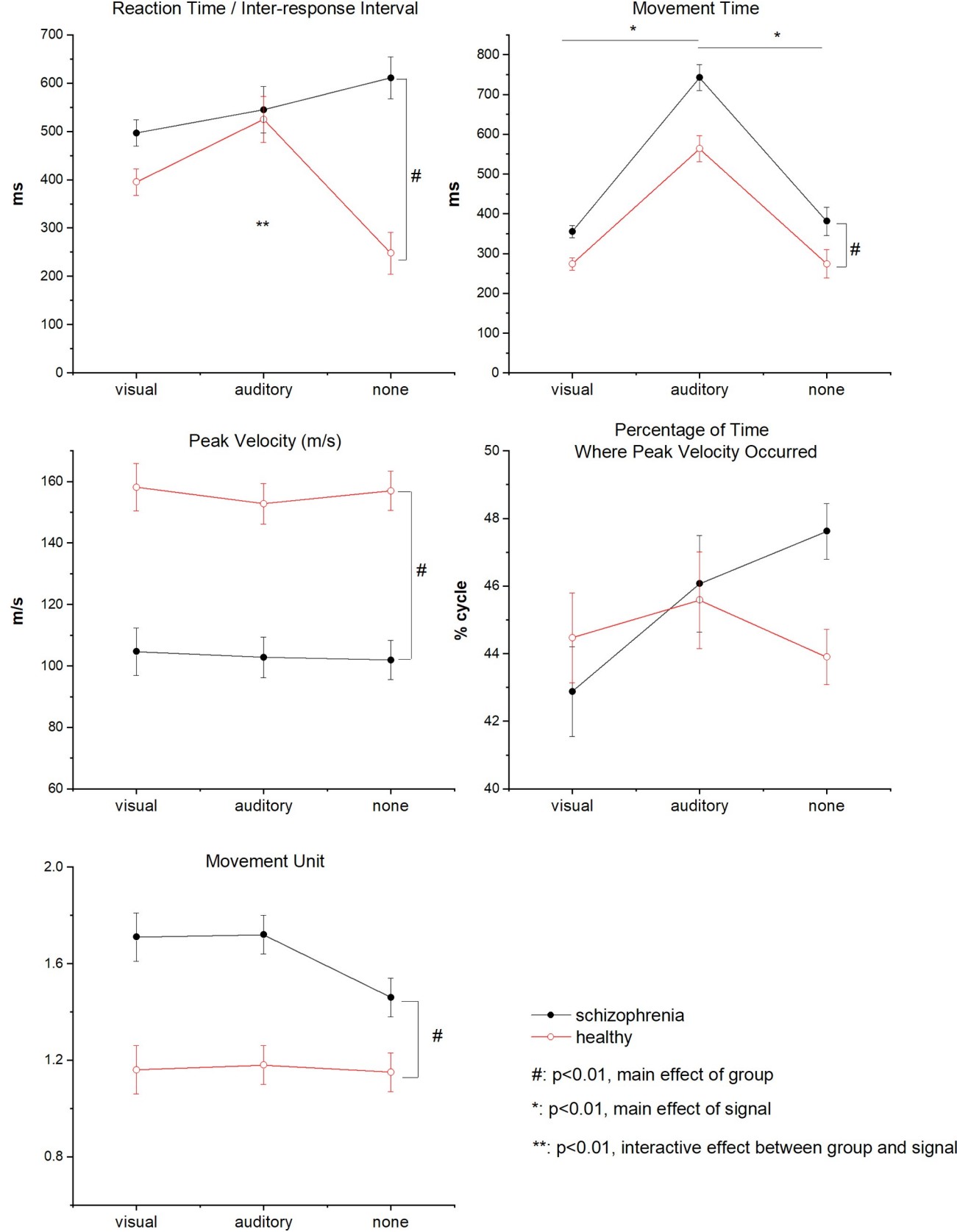

**Fig 2. Reaching performance of persons with schizophrenia and healthy control participants across three types of signal conditions.** All data were expressed as means±S.E.M.

showing that the RT of persons with schizophrenia is slower, which may be interpreted as specific psychotic symptomatology (e.g., internal distraction by hearing voices) or impaired attention [25–27]. Persons with schizophrenia respond slowly to stimuli in various RT tasks. Slower RT might be associated with sensory processing deficits [28], and it may influence persons with schizophrenia to act promptly in real-world situations. In addition, a recent study reported that persons with schizophrenia had larger intra-subject RT variability compared to that of healthy controls reflecting a deficit in information processing resulting from dysfunction of the neural system [29]. As expected, the movement time of persons with schizophrenia was slower compared to that of healthy controls. The findings are in line with the results of previous studies showing increased movement time and prolonged motor planning and execution in schizophrenia patients [30]. This aberrant motor behavior may be linked to psychomotor slowing. The term "psychomotor" considers movement or action as its principal component while stressing the involvement of perceptual processes and cognitive control mechanisms [31]. The psychomotor slowing is determined mostly by neuropsychological assessment of the speed of fine movements [10, 32–34]. A slower performance was reported to be associated with disease-induced negative symptoms [10, 32]. Negative symptoms might be primarily due to the core pathophysiology of schizophrenia while secondary ones are due to other factors, such as antipsychotic medications or the impoverished institutional environment [35, 36]. Previous studies have shown that antipsychotic medications may either improve or deteriorate motor function [37, 38]. The exact effects of medication on persons with

**Table 2. Pearson's correlation coefficient (r) between reaching performance and both duration of illness and dosage of antipsychotics.**

| Reaching performance | Duration of illness | Dosage of antipsychotics |
|---|---|---|
| Visual signal | | |
| RT | 0.138 | 0.204 |
| MT | -0.158 | 0.148 |
| PV | 0.043 | -0.245 |
| PPV | -0.324 | -0.218 |
| MU | 0.282 | 0.196 |
| Auditory signal | | |
| RT | 0.113 | 0.441 |
| MT | 0.284 | 0.248 |
| PV | -0.189 | -0.217 |
| PPV | -0.119 | -0.418 |
| MU | 0.264 | 0.441 |
| No signal | | |
| IRI | 0.159 | 0.130 |
| MT | 0.392 | 0.443 |
| PV | -0.232 | -0.335 |
| PPV | -0.225 | 0.195 |
| MU | 0.297 | -0.099 |

RT: reaction time, MT: movement time, PV: peak velocity, PPV: percentage of time where peak velocity occurred, MU: movement unit, IRI: inter-response interval

schizophrenia are difficult to determine, which could explain the lack of correlation between reaching performance and antipsychotic doses in this research. Fuller and Jahanshahi (1999) also reported that the dose of neuroleptic medication did not significantly correlate with any of their measures of motor tasks [10].

In addition to slowed movement, poor coordination marked by less forceful and less smooth arm reaching was also observed in persons with schizophrenia. Poor coordination has been attributed to motoric neurological soft signs which occur in most persons with schizophrenia, including chronic cases and neuroleptic-naive first-episode patients [39, 40]. Neurological soft signs are subtle neurological abnormalities in motor coordination, sensory integration, primitive reflexes, and sequencing of complex motor acts [41, 42]. Neurological soft signs have been considered one of the target features [43] and a potential endophenotype for schizophrenia [44]. Previous studies have reported that rehabilitation interventions in schizophrenia include cognitive remediation, psychoeducation, and cognitive-behavioral therapy [45]. Considering that persons with schizophrenia may have movement disorder, movement and coordination training may be incorporated into schizophrenia rehabilitation.

The faster the stimulus reaches the brain, the faster the signal is processed. Accordingly, the reaction is faster. Since the auditory stimulus reaches the cortex faster than the visual stimulus, the auditory reaction time is faster compared to the visual reaction time. In a previous study, simple RT (one stimulus and one response) was 160ms for sound stimulus and 190ms for light stimulus in college-age individuals [46]. It means that the auditory stimulus would be detected and processed faster compared to a visual stimulus when the stimulus is presented separately. Shelton and Kumar also showed that the simple auditory RT is faster compared to the visual RT [47]. However, when both stimuli were delivered simultaneously, subjects responded predominantly to the light, unaware of the tone presented [48].

In contrast to the previous studies, RTs to auditory and visual signals did not differ significantly in our study. However, the visual and auditory signals were not presented simultaneously in our study; instead, signals were presented randomly from one of the seven bottles rather than from a single source or measured as a simple RT. Nevertheless, for no signal condition, an interaction effect was observed between group and signal. The IRI for no signal (self-initiated movement) was significantly faster compared to the RT for auditory and visual signals in healthy controls. For no signal condition in our study, the first round of arm reaching was in response to the examiner's command, whereas the following six rounds of arm reaching were self-initiated immediately after the participants' hands touched the button (end of the preceding reaching task). Studies conducted in monkeys have shown that self-initiated movement is associated with medially located supplementary motor area (SMA), and externally triggered movement is related more to the lateral premotor area [49]. The studies in humans also indicated that different cortical areas are probably involved in the generation of self-initiated and externally triggered movements [50]. However, subsequent research has suggested that self-initiated and externally triggered movements can activate the pre-SMA, SMA, and rostral cingulate cortex similarly, although the timing of the hemodynamic response within the pre-SMA is earlier for self-initiated movements compared to externally triggered movements. This reflects the pre-SMA involvement in the early stages of voluntary movement preparation [51]. On the contrary, the IRI of self-initiated movement in persons with schizophrenia was significantly slower compared to the RT of auditory and visual signals in our study. This is consistent with the previous study in which patients with schizophrenia, particularly those with negative signs, showed impairment of willed actions with lower movement-related potentials before self-initiated movements but did not show impairment of externally triggered movements [52]. Recent neuroimaging studies have provided some evidence of brain structural alteration in SMA possibly supporting the assumptions of impaired self-initiated movements in

schizophrenia patients with motor symptoms [53, 54]. An inability to initiate and persist in goal-directed activities (avolition) is included in the definition of schizophrenia in DSM-5.

Compared to visual or no signal conditions, auditory signals induced a slower MT of reaching. The subjects spent more time detecting which bottle elicited the auditory signal because the signal was presented at one of the seven cylindrical bottles. We assumed that it is more difficult to detect the source of sound than that of light when the signal comes from the same spatial location as the other competing sources [55]. Furthermore, the previous study found that persons with schizophrenia had decreased sensitivity to auditory detection and discrimination [56]. Additionally, persons with schizophrenia showed a significant visual over auditory preference, committing more auditory commission and omission errors compared to visual errors in the simple modal condition [57]. Therefore, a visual signal or cue should be provided to persons with schizophrenia to enhance their reaching performance.

For a single-peaked velocity profile of reaching, the duration of the deceleration phase increased (a lower PPV) together with accuracy demands because of increased feedback control [58]. The results of this study showed that the PPV did not vary between two groups and three types of signal conditions, indicating no between-group differences in the motor control strategy during arm reaching. The accuracy demand of the reaching target in our study did not seem to be different for persons with schizophrenia and healthy controls, and the accuracy demand also did not differ across signal conditions.

One limitation of this study was that the sample size was relatively small. However, the variability of those measured parameters was not high (all the coefficients of variation were less than 50%, ranging from 7.18% to 47.08%). Therefore, 20 patients with 20 healthy controls should have been enough to investigate the kinematic differences in reaching performance. In addition, we enrolled healthy controls instead of other chronic psychotic patients because of this study aimed to examine the differences between persons with schizophrenia and healthy subjects. Further research may include participants with other types of mental illness and compare their reaching performance to that of schizophrenia patients.

As mentioned, we recruited only persons with chronic schizophrenia, which is also another limitation of this study. Thereby, the findings may not be generalizable to all persons with schizophrenia. In addition, we did not assess the current symptoms of the patients that may affect motor behavior measures. Future research may replicate this study with patients at different stages of recovery and examine whether the symptoms observed are associated with reaching performance.

## Conclusions

Reaching performance was poor in persons with chronic schizophrenia, as demonstrated by slower, less forceful, and less smooth movement. In addition, in persons with chronic schizophrenia, the inter-response interval for the self-initiated movement was the longest and movement time of arm reaching was longer when guided by the auditory (sound) signal compared to the visual (light) signal. Thus, further research might incorporate visual signals to examine the effects of movement coordination training on reaching performance.

## Supporting information

**S1 Dataset.**
(PDF)

**S2 Dataset.**
(XLSX)

**S1 Data.**
(XLSX)

# Acknowledgments

All the participants are gratefully acknowledged.

# Author Contributions

**Conceptualization:** Paul Sin-Bao Huang, Shu-Zon Lou.

**Data curation:** Paul Sin-Bao Huang, Kwok-Tak Yeung, Ming-Yi Hsu, Sok-Wa Wan.

**Formal analysis:** Sok-Wa Wan.

**Funding acquisition:** Paul Sin-Bao Huang.

**Investigation:** Kwok-Tak Yeung, Ming-Yi Hsu, Sok-Wa Wan.

**Methodology:** Chiung-Ling Chen, Ming-Yi Hsu.

**Writing – original draft:** Paul Sin-Bao Huang, Chiung-Ling Chen.

**Writing – review & editing:** Shu-Zon Lou.

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
