## [Decision Letter · Decision Letter 0]

2 Jan 2020

PONE-D-19-26482

Effects of different types of sensory signals on reaching performance in persons with chronic schizophrenia

PLOS ONE

Dear Dr. Lou,

Thank you for submitting your manuscript to PLOS ONE. After careful consideration, we feel that it has merit but does not fully meet PLOS ONE’s publication criteria as it currently stands. Therefore, we invite you to submit a revised version of the manuscript that addresses the points raised during the review process.

We would appreciate receiving your revised manuscript by Feb 14 2020 11:59PM. To enhance the reproducibility of your results, we recommend that if applicable you deposit your laboratory protocols in protocols.io, where a protocol can be assigned its own identifier (DOI) such that it can be cited independently in the future. For instructions see: http://journals.plos.org/plosone/s/submission-guidelines#loc-laboratory-protocols

We look forward to receiving your revised manuscript.

Kind regards,

Robin Baurès, Ph.D.

Academic Editor

PLOS ONE

Additional Editor Comments:

Dear Shu-Zon Lou,

I have now received two reviews regarding your manuscript. As you will see, the two reviewers have a different opinion on your manuscript. R1 is quite convinced with your experiment and results, still suggesting some improvement or raising important questions you would need to address. R2 on the contrary raises important, major concerns that might definitely be harder to address. I have read myself the paper, and I am inclined to give you an opportunity to revise your manuscript. As this might be hard to address all these comments, the questions raised are definitely important to make sure your data truly support the results you claim.

In addition, your manuscript has raised some methodological warning during its internal evaluation (from the journal staff). I copy here the exact exchange we had :

Initial remark received:

Note from Staff Editor : During our internal evaluation of this manuscript, we noted that the sample size involved in this study may not be sufficient to considering the conclusions drawn. Additionally, the control group may not be appropriate. I would be grateful if you could pay particular attention to these aspects during your initial evaluation of the manuscript.

My answer:

I had a look to this part before sending for review, and did not spot a particular issue there. With 20 participants in each group, this is much more than what is done in my own field, hence that seemed fine to me. Could you please tell me where lays the issue here? Does it mean 20 is not enough, or 20 patients with schizophrenia as it is a particular and heterogeneous disease?

The final reply:

The issues raised concerned both the heterogeneity of the condition and the nature of the control group - i.e. the use of healthy relatives of staff rather than a cohort of patients lacking the proposed sensory symptoms. This would act to minimise potential confounding factors. However, I am content for this manuscript to go out for peer review if you think the control group and sample size are appropriate.

As you can see, as I agree with the methodology you used, this is not going without questions from other readers. I would like you to take these comments into account, and insert in the discussion a paragraph mentioning the choice you made and the alternative method, the pro and cons of each.

I hope you will be able to answer each of the comments that have been formulated by the reviewers, and do thank you for considering Plos One for publishing your work.

Best,

Robin Baurès

Journal Requirements:

2. Please describe in your methods section how capacity to consent was determined for the participants in this study.

3. We note that Figure 1 includes an image of a participant in the study. 

4.  Your ethics statement must appear in the Methods section of your manuscript. If your ethics statement is written in any section besides the Methods, please move it to the Methods section and delete it from any other section. Please also ensure that your ethics statement is included in your manuscript, as the ethics section of your online submission will not be published alongside your manuscript.

Reviewers' comments:

Reviewer's Responses to Questions

**Comments to the Author**

1. Is the manuscript technically sound, and do the data support the conclusions?

Reviewer #1: Yes

Reviewer #2: No

2. Has the statistical analysis been performed appropriately and rigorously? 

Reviewer #1: Yes

Reviewer #2: Yes

3. Have the authors made all data underlying the findings in their manuscript fully available?

Reviewer #1: Yes

Reviewer #2: Yes

4. Is the manuscript presented in an intelligible fashion and written in standard English?

Reviewer #1: Yes

Reviewer #2: No

5. Review Comments to the Author

Reviewer #1: this is a very interesting and timely study on reaching movement in schizophrenia patients. Authors applied a novel technique to objectively assess motor behavior during a reaching task. They find that reaction times, movement times and amount of movement is increased in patients, while speed is lower than in healthy controls. The results fit nicely in the recent discussion on specific motor deficits in psychotic disorders. Overall the manuscript is well-written. I have a few suggestions for further improvement:

1. line 56 typo in era

2. I wonder how authors dealt with the repetitions in their statistical analysis. to my opinion they should adopt a repeated measures design or at least demonstrate that the three repetitions had no extra effects and that it is ok to average performance across these trials.

3. one drawback of this study is the lack of information on current symptoms. these would have been able to link to motor behavior measures

4. The first paragraph of the discussion is lengthy - contains duplicate information. Suggestion is to use the full wording at first use in the discussion for the variables instead of abbreviations. This will help readers. for detailed definitions readers should refer to the methods section, which contains sufficient details.

5. When discussing their findings, authors are referred to Dutschke et al. 2018 Schizophrenia Research, who applied automated video analysis of gestures in schizophrenia and also found increased movement time and increased movement in patients as compared to healthy controls.

6. Authors should also discuss recent reports on SMA dysfunction/hyperactivity/hyperconnectivity in schizophrenia that was also linked to motor abnormalities in these patients. There are neuroimaging findings that already support the assumptions by the authors

7. please test whether current antipsychotic dosage or duration of illness was linked to the movement parameters.

8. please consider using line graphs instead of bars (horizontal axis would depict the type of stimulus and lines with error bars would indicate the groups).

9. discussion, limitation section: lack of cognitive impairment cannot be assumed from MMSE scores > 28. Typically, chronic schizophrenia is associated with distinct alterations of cognitive domains that are more subtle than the MMSE.

Reviewer #2: This study investigates reaching movement behaviour in three tasks in a group of 20 patients with chronic schizophrenia and an aged matched control group. The authors report differences in reaction time (patients being more slow than healthy controls) as well as in the reaching movements per se (larger movement times, lower maximum velocity of the reaching movements and larger number of zero crossings in the velocity profile of the reaching movements). The authors suggest that these deviances lead to the conclusion that movement training of patients can benefit from the use of visual signals.

There are several major issues with this study both in the design of the study and the interpretation of the results.

1. It is well known that movement behaviour is affected in patients with psychosis taking anti-dopaminergic medications by the side effects of these medications in the motor system including slower movements, tremor and a reduction the maximum velocity of the movements. All these effects could be attributed to parkinsonian like symptoms due to the action of the anti-dopaminergic medications. This study uses chronic patients that presumably receive such medications. The authors don’t provide any information on the patient medication status and this a crucial parameter here. The medication status could probably explain all of these effects on reaching performance and the authors don’t even mention this possibility when discussing their results, attributing differences in movement parameters to negative symptoms and neurological soft signs. In fact neither negative symptoms nor neurological soft signs were measured in this study to provide any evidence of correlation with the movement deviances observed in the patients. The most obvious reason though for movement slowing and movement break down in patients would be the effect of antipsychotic medications. I would also expect that these effects would correlate with the doses of these medications as well as the time that the patients received the medications.

2. The design of the study has a very severe methodological problem. The authors use a paradigm where subjects either receive a visual or auditory signal instructing them to move to a target in 3D space (cylinder in front of them) or perform self-initiated movements to the same targets in a predefined sequence. The authors measure reaction times in all three tasks and compare these among the three tasks. Importantly the authors find significant differences in RT among patients and controls specific to the task. The major problem here is that one cannot measure reaction time in self-paced movements because the definition of reaction time requires an external event triggering the movement. What the authors measure in the self-paced task is the internal timing of the subject as he/she proceeds to move back and forth from one stimulus to the next. This is not reaction time but an internal self- paced rhythm. It is known that parkinsonism affects this self-paced rhythm of sequential movements so the fact that patients were particularly slow in initiating and performing this sequence could very well be again the result of anti-dopaminergic medication. In any case one cannot compare RT in externally triggered movements with what is erroneously called by the authors RT in self-paced movements because RT (REACTION TIME) is defined ONLY for externally triggered responses. The analysis then of RT should be confined to externally triggered (auditory and visual) movements. The measuring of the time from the initiation of one self-paced movement to the next is a different measure NOT REACTION TIME and should be treated separately. Any conclusions drown on this measure should also be treated separately as indications of an internally generated rhythmic movement pattern and NOT RT. Since the authors’ main conclusion is based on a difference in this ill- defined measure I believe that the argument they build on is also erroneous (see my specific comments on discussion).

3. The difference in movement time for the auditory condition as the authors also point out in the discussion is probably due to the difficulty of the subjects in accurately locating the target in space based on an auditory signal. This is a very well-known effect in perception namely that locating in space auditory targets is much less efficient than locating visual targets. Thus this difference is not related to movement per se but to perception. My guess is that subjects started the movement by releasing the start button as soon as they heard the stimulus which was as fast as the visual condition but then proceeded slowly in order to accumulate more information on the exact source of the sound and reach the appropriate target. Thus they used a different strategy for reaching to the auditory targets that was the result of the difficulty in locating the source of the auditory signal in space. In my opinion then one cannot directly compare reaching performance in the auditory with the visual and self- paced target conditions since the former also involves the added perceptual task difficulty of accurately locating the auditory source of the target in space.

4. The conclusion of the study that movement coordination training is the cornerstone of comprehensive treatment in patients with schizophrenia is completely irrelevant to the design and results of the study. How do the results of this study relate to this conclusion? Did the authors test the effects of such “movement training” as treatment for schizophrenia? Schizophrenia is a mental disorder mainly affecting thought and perception. A movement training therapy is the “corner stone” of comprehensive treatment in schizophrenia? If that is so then we should all refer our psychotic patients to movement rehabilitation programs!!!!!

Specific Comments

Abstract

1. The abstract is badly written and is very confusing to the reader. For example the sentence: “The persons with chronic schizophrenia demonstrated much more reaction time, movement time, and movement units but less peak velocity” is incomprehensible. The reaction time was increased in patients with schizophrenia or patients were slower to respond. What is movement unit? The authors refer here to acceleration zero crossings that suggest a breakdown of the movement smoothness but the non-expert reader cannot understand this.

2. “The reaction time of reaching was shortest for no signal condition in healthy controls.”

This is absurd. How can one react to a NO SIGNAL? See my second comment.

3. The conclusion of the abstract is completely irrelevant to this study.

Introduction

1. Line 83: this paragraph discusses methods of rehabilitation in people with stroke and hemiplegia that have severe problems in reaching. How is this related with schizophrenia? Schizophrenia is not a movement disorder. Although there are some subtle changes, the soft neurological signs, in some of the patients these do not constitute a major feature of the disorder. Also “psychomotor slowing” refers more to a slowing of processing leading to action in patients with schizophrenia compared to controls as for example in measuring reaction time in sensory-motor tasks. Thus the emphasis here is on cognitive processes leading to a slowing of movements and not a movement deficit per se as observed in hemiplegia resulting from stroke. Applying movement rehabilitation programs in patients with schizophrenia is probably not going to work because their deficit is in higher cognition rather than motor control. In any case this study does not address this issue at all so I think this whole discussion is irrelevant here.

Methods

1. The task as described by the authors involves 2 reaching movements in each trial, one reaching to the target cylinder and one returning to the start button. Do the authors measure separately the movement characteristics of the outgoing and return movement? If not then they should. Also the movement units are measured separately for the outgoing and returning movement? They should since the subject has to make two distinct movements in this design, one to reach the target cylinder and a second one to return to the start button. Also it is not clear to me what the instruction was to the subject. Was it to reach and touch the cylinder and then return home, or to grab the cylinder? In any case the trial involves definitely TWO reaching movements and any measures should be clearly attributed to either one or the other of the two movements. The authors define movement time as the time from release of start button to the time of press thus including both reaching movements. Later though they define reaching based on velocity as the point where velocity was above 3% and then below 3% of its peak when the hand touched the bottle. So which was the movement? The single movement defined by the velocity or the two movement sequence defined by the movement time?

Discussion

1. The increase of RT in schizophrenia has been observed in a large body of literature (Nuechterlein 1977). The authors could use more recent literature on this phenomenon implicating decision processes rather than attention (Karantinos et al 2014).

2. Does this study proves why movement coordination therapy is needed for patient with schizophrenia? Do the authors show here any relevant results?

3. Why the earlier timing of hemodynamic response in SMA proves that self-initiated movements are more efficient compared to externally triggered movements? What do the authors mean by more efficient?

4. The authors mention a study in which they say that self-initiated movements were impaired in patients with schizophrenia. They study though claims that motor potentials prior to self-initiated movements were impaired in patients. Also there is no consistent evidence that self-initiated movements are impaired in these patients. In any case what do we mean here by impaired?

5. The authors claim that their patients did not have cognitive impairment. How do they make this claim? The use of the MMPI is not enough to claim that a patient with schizophrenia has no cognitive impairment. The need for specific instruments designed to assess cognitive dysfunction in schizophrenia is needed here (for example the MATRICS etc). It is known that these patients have specific problems in executive function, working memory and speed of performance that are not assessed by the MMPI.

6. PLOS authors have the option to publish the peer review history of their article (what does this mean?). If published, this will include your full peer review and any attached files.

Reviewer #1: No

Reviewer #2: Yes: Nikolaos Smyrnis

---

## [Author Response · Author response to Decision Letter 0]

5 Mar 2020

Date: March 1, 2020

To: Robin Baurès, Ph.D.,.

Manuscript ID number: PONE-D-19-26482

Dear Dr. Baurès:

Thank you for giving us the opportunity to revise our manuscript entitled “Effects of different types of sensory signals on reaching performance in persons with chronic schizophrenia” (ID numberPONE-D-19-26482). We have revised the manuscript as suggested by the reviewers. In addition, we have inserted in the discussion a paragraph mentioning the sample size and the choice of controls as you suggested. Please see the list of the revisions in the attached Detailed Response to Reviewers.

Accordingly, we have uploaded a copy of highlighted version and a copy of clean version of the revised manuscript. The revised manuscript has been edited by a skilled writer of English.

We would like to take this opportunity to express our sincere thanks to the reviewers for giving us their kind advice and valuable comments. We sincerely hope that our manuscript could be accepted for publication in your prestigious journal.

Sincerely Yours,

Shu-Zon Lou, Ph. D

Assistant Professor,

Department of Occupational Therapy, Chung Shan Medical University

Detailed Response to Reviewers

Manuscript ID number: PONE-D-19-26482

Responses to Reviewer #1:

Thank you so much for giving us your kind advice and valuable comments. We have revised this paper according to your suggestions. Please see the following point-by-point responses to your comments. Your comments are quoted in black font, our responses are in blue font and all the changes made in the revised manuscript are marked using the Microsoft Word's Track changes. The page and line numbers are in accordance with the highlighted version.

This is a very interesting and timely study on reaching movement in schizophrenia patients. Authors applied a novel technique to objectively assess motor behavior during a reaching task. They find that reaction times, movement times and amount of movement is increased in patients, while speed is lower than in healthy controls. The results fit nicely in the recent discussion on specific motor deficits in psychotic disorders. Overall the manuscript is well-written. I have a few suggestions for further improvement:

1. line 56 typo in era

[Reply]: corrected as suggested, Line 63

2. I wonder how authors dealt with the repetitions in their statistical analysis. to my opinion they should adopt a repeated measures design or at least demonstrate that the three repetitions had no extra effects and that it is ok to average performance across these trials.

[Reply]: Repeated measures design involves multiple measurements of the same variable taken on the same or matched subjects either under different conditions or over two or more time periods. Repeated measurements are collected in a longitudinal study in which change over time is assessed. However, in this study we collected cross-sectional data and measured the performance at a specific point in time. We thought that an average of three trials are necessary to provide stable data in reaching task (Chen, Garcia-Vergara & Howard, 2019).

Chen, Y., Garcia-Vergara, S., & Howard, A. (2019). Number of trials necessary to achieve performance stability in a reaching kinematics movement analysis game. J Hand Ther. doi:10.1016/j.jht.2019.04.001

3. one drawback of this study is the lack of information on current symptoms. these would have been able to link to motor behavior measures

[Reply]: “lack of information on current symptoms” has been written as limitation in the “Discussion” section. 

Line 407-410, “A further limitation of this study was that we recruited only persons with chronic schizophrenia. Thereby, the findings may not be generalizable to all persons with schizophrenia. In addition, we did not collect the current symptoms of the patients that may affect motor behavior measures.”

4. The first paragraph of the discussion is lengthy - contains duplicate information. Suggestion is to use the full wording at first use in the discussion for the variables instead of abbreviations. This will help readers for detailed definitions readers should refer to the methods section, which contains sufficient details.

[Reply]: The first paragraph has been deleted. Some detailed description about measurement of reaching performance has been added to the “Methods” section. 

Line 168-190

5. When discussing their findings, authors are referred to Dutschke et al. 2018 Schizophrenia Research, who applied automated video analysis of gestures in schizophrenia and also found increased movement time and increased movement in patients as compared to healthy controls.

[Reply]: The reference has been added in the discussion.

Line 304: “The findings are in line with the results of previous studies showing increased movement and prolonged motor planning and execution in schizophrenic patients [30]. This aberrant motor behavior may be linked to psychomotor slowing.”

6. Authors should also discuss recent reports on SMA dysfunction/hyperactivity/hyperconnectivity in schizophrenia that was also linked to motor abnormalities in these patients. There are neuroimaging findings that already support the assumptions by the authors

[Reply]: Recent reports on SMA dysfunction in schizophrenia have been added in discussion. 

Line 373, “Recent neuroimaging studies have provided some evidence of brain structural alteration in SMA possibly supporting the assumptions of impaired self-initiated movements in schizophrenia patients with motor symptoms.”

7. please test whether current antipsychotic dosage or duration of illness was linked to the movement parameters.

[Reply]: The results of relationship between movement parameters and current antipsychotic dosage, and duration of illness have been added in the revised manuscript.

Line 123, “The antipsychotic doses were collected for each patient and converted to chlorpromazine equivalents (CPZE)” 

Line 233, “In addition, Pearson correlation was conducted to examine the relationship between reaching performance and clinical characteristics.”

Line 265, “ The results of Pearson correlation tests showed no significant associations between reaching performance and clinical characteristics, which were antipsychotic dosage and duration of illness (all p values > 0.01).”

8. please consider using line graphs instead of bars (horizontal axis would depict the type of stimulus and lines with error bars would indicate the groups).

[Reply]: the figure has been revised as suggested

9. discussion, limitation section: lack of cognitive impairment cannot be assumed from MMSE scores > 28. Typically, chronic schizophrenia is associated with distinct alterations of cognitive domains that are more subtle than the MMSE.

[Reply]: The description about cognitive impairment has been deleted 

Line 408

Detailed Response to Reviewers

Manuscript ID number: PONE-D-19-26482

Responses to Reviewer #2:

Thank you so much for giving us your kind advice and valuable comments. We have revised this paper according to your suggestions. Please see the following point-by-point responses to your comments. Your comments are quoted in black font, our responses are in blue font and all the changes made in the revised manuscript are marked using the Microsoft Word's Track changes. The page and line numbers are in accordance with the highlighted version.

This study investigates reaching movement behaviour in three tasks in a group of 20 patients with chronic schizophrenia and an aged matched control group. The authors report differences in reaction time (patients being more slow than healthy controls) as well as in the reaching movements per se (larger movement times, lower maximum velocity of the reaching movements and larger number of zero crossings in the velocity profile of the reaching movements). The authors suggest that these deviances lead to the conclusion that movement training of patients can benefit from the use of visual signals.

There are several major issues with this study both in the design of the study and the interpretation of the results.

1. It is well known that movement behaviour is affected in patients with psychosis taking anti-dopaminergic medications by the side effects of these medications in the motor system including slower movements, tremor and a reduction the maximum velocity of the movements. All these effects could be attributed to parkinsonian like symptoms due to the action of the anti-dopaminergic medications. This study uses chronic patients that presumably receive such medications. The authors don’t provide any information on the patient medication status and this a crucial parameter here. The medication status could probably explain all of these effects on reaching performance and the authors don’t even mention this possibility when discussing their results, attributing differences in movement parameters to negative symptoms and neurological soft signs. In fact neither negative symptoms nor neurological soft signs were measured in this study to provide any evidence of correlation with the movement deviances observed in the patients. The most obvious reason though for movement slowing and movement break down in patients would be the effect of antipsychotic medications. I would also expect that these effects would correlate with the doses of these medications as well as the time that the patients received the medications.

[Reply]: The antipsychotic doses has been collected for each patient and converted to chlorpromazine equivalents (CPZE) 

Line 123, “The antipsychotic doses were collected for each patient and converted to chlorpromazine equivalents (CPZE)”

The relationship between reaching performance and doses of medications has been tested. 

Line 233, “In addition, Pearson correlation was conducted to examine the relationship between reaching performance and clinical characteristics.” 

Line 265, “ The results of Pearson correlation tests showed no significant associations between reaching performance and clinical characteristics, which were antipsychotic dosage and duration of illness (all p values > 0.01).”

Discussion about the result was added in “Discussion” section. 

Line 315-319” Previous studies have shown that antipsychotic medications may either improve or deteriorate motor function [37, 38]. The exact effects of medication on persons with schizophrenia are difficult to determine, which could explain the lack of correlation between reaching performance and antipsychotic doses in this research.” 

2. The design of the study has a very severe methodological problem. The authors use a paradigm where subjects either receive a visual or auditory signal instructing them to move to a target in 3D space (cylinder in front of them) or perform self-initiated movements to the same targets in a predefined sequence. The authors measure reaction times in all three tasks and compare these among the three tasks. Importantly the authors find significant differences in RT among patients and controls specific to the task. The major problem here is that one cannot measure reaction time in self-paced movements because the definition of reaction time requires an external event triggering the movement. What the authors measure in the self-paced task is the internal timing of the subject as he/she proceeds to move back and forth from one stimulus to the next. This is not reaction time but an internal self- paced rhythm. It is known that parkinsonism affects this self-paced rhythm of sequential movements so the fact that patients were particularly slow in initiating and performing this sequence could very well be again the result of anti-dopaminergic medication. In any case one cannot compare RT in externally triggered movements with what is erroneously called by the authors RT in self-paced movements because RT (REACTION TIME) is defined ONLY for externally triggered responses. The analysis then of RT should be confined to externally triggered (auditory and visual) movements. The measuring of the time from the initiation of one self-paced movement to the next is a different measure NOT REACTION TIME and should be treated separately. Any conclusions drown on this measure should also be treated separately as indications of an internally generated rhythmic movement pattern and NOT RT. Since the authors’ main conclusion is based on a difference in this ill- defined measure I believe that the argument they build on is also erroneous (see my specific comments on discussion).

[Reply]: the term RT for no signal has been changed to Inter-Response Interval (IRI) for self-initiated movement.

3. The difference in movement time for the auditory condition as the authors also point out in the discussion is probably due to the difficulty of the subjects in accurately locating the target in space based on an auditory signal. This is a very well-known effect in perception namely that locating in space auditory targets is much less efficient than locating visual targets. Thus this difference is not related to movement per se but to perception. My guess is that subjects started the movement by releasing the start button as soon as they heard the stimulus which was as fast as the visual condition but then proceeded slowly in order to accumulate more information on the exact source of the sound and reach the appropriate target. Thus they used a different strategy for reaching to the auditory targets that was the result of the difficulty in locating the source of the auditory signal in space. In my opinion then one cannot directly compare reaching performance in the auditory with the visual and self- paced target conditions since the former also involves the added perceptual task difficulty of accurately locating the auditory source of the target in space.

[Reply]: The term reaching performance is used to describe what is seen during testing of the motor task. In clinical practice, it is unusual to find motor task that does not have a cognitive component or the reverse. Therefore, we considered the function involved in reaching task may refer to as psychomotor function which includes basic motor skill and cognitive process. 

There were some studies that compared reaction time for visual and auditory signals (Chan et. al., 2006, Shelton, et al., 2010) in literature. There were also studies compared self-initiated and externally triggered movements (Cunnington, et al., 2002, Jenkins, et. al., 2000). We don’t think it was inappropriate to directly compare reaching performance in the auditory with the visual and self- paced target conditions.

4. The conclusion of the study that movement coordination training is the cornerstone of comprehensive treatment in patients with schizophrenia is completely irrelevant to the design and results of the study. How do the results of this study relate to this conclusion? Did the authors test the effects of such “movement training” as treatment for schizophrenia? Schizophrenia is a mental disorder mainly affecting thought and perception. A movement training therapy is the “corner stone” of comprehensive treatment in schizophrenia? If that is so then we should all refer our psychotic patients to movement rehabilitation programs!!!!!

[Reply]: In discussion section, we have rephrased the clinical implication of our results, and changed to a more cautious suggestion regarding coordination training. The effects of movement coordination were suggested as further research. 

Line332, “Considering that persons with schizophrenia may have movement disorder, movement and coordination training may be incorporated into schizophrenia rehabilitation.”

Line423, “Thus, further research might incorporate visual signals to examine the effects of movement coordination training on reaching performance. ”

Specific Comments

Abstract

1. The abstract is badly written and is very confusing to the reader. For example the sentence: “The persons with chronic schizophrenia demonstrated much more reaction time, movement time, and movement units but less peak velocity” is incomprehensible. The reaction time was increased in patients with schizophrenia or patients were slower to respond. What is movement unit? The authors refer here to acceleration zero crossings that suggest a breakdown of the movement smoothness but the non-expert reader cannot understand this.

[Reply]: Line 35: The description about reaching performance has changed to “slower response to signals and self-initiated movement, increased movement time, less forceful and less smooth movement” for easily understood. 

2. “The reaction time of reaching was shortest for no signal condition in healthy controls.”

[Reply]: Line 40: the description has changed to “the inter-response interval of self-initiated reaching was shortest in healthy controls.”

This is absurd. How can one react to a NO SIGNAL? See my second comment.

3. The conclusion of the abstract is completely irrelevant to this study.

[Reply]: reaction time to no signal has been change to inter-response interval for self-initiated movement.

The irrelevant description in conclusion has been deleted

Introduction

1. Line 83: this paragraph discusses methods of rehabilitation in people with stroke and hemiplegia that have severe problems in reaching. How is this related with schizophrenia? Schizophrenia is not a movement disorder. Although there are some subtle changes, the soft neurological signs, in some of the patients these do not constitute a major feature of the disorder. Also “psychomotor slowing” refers more to a slowing of processing leading to action in patients with schizophrenia compared to controls as for example in measuring reaction time in sensory-motor tasks. Thus the emphasis here is on cognitive processes leading to a slowing of movements and not a movement deficit per se as observed in hemiplegia resulting from stroke. Applying movement rehabilitation programs in patients with schizophrenia is probably not going to work because their deficit is in higher cognition rather than motor control. In any case this study does not address this issue at all so I think this whole discussion is irrelevant here.

[Reply]: The description about rehabilitation in people with stroke cited from literature has been deleted. Line 93-96

Movement rehabilitation programs focus on motor learning which is a set of internal processes associated with practice or experience leading to relatively permanent changes in the capability for skilled behavior. Therefore, movement rehabilitation involves both cognitive processing and motor skill training not only movement per se.

Methods

1. The task as described by the authors involves 2 reaching movements in each trial, one reaching to the target cylinder and one returning to the start button. Do the authors measure separately the movement characteristics of the outgoing and return movement? If not then they should. Also the movement units are measured separately for the outgoing and returning movement? They should since the subject has to make two distinct movements in this design, one to reach the target cylinder and a second one to return to the start button. Also it is not clear to me what the instruction was to the subject. Was it to reach and touch the cylinder and then return home, or to grab the cylinder? In any case the trial involves definitely TWO reaching movements and any measures should be clearly attributed to either one or the other of the two movements. The authors define movement time as the time from release of start button to the time of press thus including both reaching movements. Later though they define reaching based on velocity as the point where velocity was above 3% and then below 3% of its peak when the hand touched the bottle. So which was the movement? The single movement defined by the velocity or the two movement sequence defined by the movement time?

[Reply]: There were two ways to measure reaching performance: RT/IRI and MT of total movement (reaching out to touch and return) were measured by Interactive Sound and Light Eye-Hand Coordination Training System, and the other velocity parameters of reaching out (PV, PPV and MU) were measured by Zebris. 

Discussion

1. The increase of RT in schizophrenia has been observed in a large body of literature (Nuechterlein 1977). The authors could use more recent literature on this phenomenon implicating decision processes rather than attention (Karantinos et al 2014).

[Reply]: The more recent literature (Karantinos et al 2014) has been cited in discussion. 

Line 300, “In addition, a recent study reported that persons with schizophrenia had larger intra-subject RT variability compared to that of healthy controls reflecting a deficit in information processing resulting from dysfunction of the neural system [29].”

2. Does this study proves why movement coordination therapy is needed for patient with schizophrenia? Do the authors show here any relevant results?

[Reply]: The description about movement coordination has been rephrased to a more cautious suggestion and the “effects of movement coordination training” has been listed as further research suggestion.

3. Why the earlier timing of hemodynamic response in SMA proves that self-initiated movements are more efficient compared to externally triggered movements? What do the authors mean by more efficient?

[Reply]: Line 364, the description has been changed to “This reflects the pre-SMA involvement in the early stages of voluntary movement preparation [51].” for clarity. 

4. The authors mention a study in which they say that self-initiated movements were impaired in patients with schizophrenia. They study though claims that motor potentials prior to self-initiated movements were impaired in patients. Also there is no consistent evidence that self-initiated movements are impaired in these patients. In any case what do we mean here by impaired?

[Reply]: Line 369, the description has been changed to “This is consistent with the previous study in which patients with schizophrenia, particularly those with negative signs, showed impairment of willed actions with lower movement-related potentials before self-initiated movements but did not show impairment of externally triggered movements [52]. 

5. The authors claim that their patients did not have cognitive impairment. How do they make this claim? The use of the MMPI is not enough to claim that a patient with schizophrenia has no cognitive impairment. The need for specific instruments designed to assess cognitive dysfunction in schizophrenia is needed here (for example the MATRICS etc). It is known that these patients have specific problems in executive function, working memory and speed of performance that are not assessed by the MMPI.

 [Reply]: The description about cognitive impairment has been deleted

---

## [Decision Letter · Decision Letter 1]

2 Apr 2020

PONE-D-19-26482R1

Effects of different types of sensory signals on reaching performance in persons with chronic schizophrenia

PLOS ONE

Dear Dr. Lou,

Thank you for submitting your manuscript to PLOS ONE. After careful consideration, we feel that it has merit but does not fully meet PLOS ONE’s publication criteria as it currently stands. Therefore, we invite you to submit a revised version of the manuscript that addresses the points raised during the review process.

As you will read, R1 still has a few minor concerns that need just a little more work. This is however not the case of R2 who considers that you did not answer fully his comments, in particular his major concern. I tend to agree with him, and would like you to carefully read and answer these comments. If you would disagree with one of these comments, please specifically mention it and argue in favor of your decision. I really need to see all the comments addressed or your rationale for sticking to your point of view.

I hope you will be able to address these remaining comments and do thank you for considering Plos One for publishing your work.

We would appreciate receiving your revised manuscript by May 17 2020 11:59PM. To enhance the reproducibility of your results, we recommend that if applicable you deposit your laboratory protocols in protocols.io, where a protocol can be assigned its own identifier (DOI) such that it can be cited independently in the future. For instructions see: http://journals.plos.org/plosone/s/submission-guidelines#loc-laboratory-protocols

We look forward to receiving your revised manuscript.

Kind regards,

Robin Baurès, Ph.D.

Academic Editor

PLOS ONE

Additional Editor Comments (if provided):

Dear Dr. Shu-Zon Lou,

I have now received two reviews regarding your manuscript. As you will read, R1 still has a few minor concerns that need just a little more work. This is however not the case of R2 who considers that you did not answer fully his comments, in particular his major concern. I tend to agree with him, and would like you to carefully read and answer these comments. If you would disagree with one of these comments, please specifically mention it and argue in favor of your decision. I really need to see all the comments addressed or your rationale for sticking to your point of view.

I hope you will be able to address these remaining comments and do thank you for considering Plos One for publishing your work.

Best,

Robin Baurès

Reviewers' comments:

Reviewer's Responses to Questions

**Comments to the Author**

1. If the authors have adequately addressed your comments raised in a previous round of review and you feel that this manuscript is now acceptable for publication, you may indicate that here to bypass the “Comments to the Author” section, enter your conflict of interest statement in the “Confidential to Editor” section, and submit your "Accept" recommendation.

Reviewer #1: All comments have been addressed

Reviewer #2: (No Response)

2. Is the manuscript technically sound, and do the data support the conclusions?

Reviewer #1: Yes

Reviewer #2: No

3. Has the statistical analysis been performed appropriately and rigorously? 

Reviewer #1: I Don't Know

Reviewer #2: Yes

4. Have the authors made all data underlying the findings in their manuscript fully available?

Reviewer #1: (No Response)

Reviewer #2: Yes

5. Is the manuscript presented in an intelligible fashion and written in standard English?

Reviewer #1: Yes

Reviewer #2: No

6. Review Comments to the Author

Reviewer #1: a few minor issues remain to be resolved:

abstract: final sentence is misleading. Authors did not test whether motor performance was linked to clinical status, except for CPZ and DOI. Please omit this information in the abstract

table 1 Levels of antipsychotics should be "dosage of antipsychotics" because authors do not report blood levels

results: again please delete the phrase on clinical characteristics (correlation with motor), but instead state that DOI and CPZ did not correlate (line 265 ff)

discussion and throughout manuscript: please replace "schizophrenic patients" with "schizophrenia patients" or "patients with schizophrenia". Schizophrenic is considered to be a stigmatizing adjective

Reviewer #2: In this revised manuscript the authors have taken into consideration some of my previous comments but they did not address adequately several others and especially my criticism which is presented in comment number 3.

As I pointed out one cannot directly compare the time interval between successive movements in a self-paced sequence movement task which is the no signal condition with the externally triggered visual and auditory task in which the time of onset for each response is the reaction time. The authors just renamed RT in the no signal task to IRI. What they should have done is a separate analysis of RT in the visual and auditory conditions comparing patients and controls and a separate analysis for IRI in the self-paced movement task comparing the IRI between patients and controls. The discussion of these two measures should also be separate because different cognitive processes are engaged in self-paced sequential movements and different processes are engaged in externally driven RT processes. The authors do mention the relevant neural systems that are engaged in self-paced sequential movements that involve the prefrontal cortex and particularly area SMA. Indeed sequential movements are pre-programmed and executed in a unified manner. The authors observe a large difficulty of patients in initiating each component of the sequence of these sequential movements and this is very relevant to the difficulty of the patients in maintaining the sequence in working memory and initiating each movement in the sequence. This effect is different from the difficulty of the patients in initiating externally triggered movements to a target where the system involves the allocation of attention to the external stimulus and the programming of a response to this stimulus. A different fronto-parietal network is activated in these stimulus response tasks. In essence the authors observe differences between patients and controls in two different systems, one related to the programming of a sequence of movements and one to attending and responding to external stimuli. This is why I still think the authors should perform separate analyses for RT and IR and discuss the differences between patients and controls separately for RT and IRI.

IN response to my comment on the effects of medication the authors report no correlation of medication to all measured variables. Could they provide the r coefficients in a table or a supplementary table?

In response to my comment about the measurement of peak velocity and movement units there is still no clear answer. I commented in my review that the subjects perform two arm movements each time one reaching out to the target and one returning back to the home position and pressing the start button. The authors explained that the MT was the time from release of the home button to the pressing of it so MT corresponds to both arm movements. What about PV and MU? The authors say in their response that they get these measurements from the “zebris” system. That was not my question. My question was WHICH movement’s profile was used to measure these parameters? The reaching out movement or the returning movement or both?

The description of the results need rephrasing, for example the phrase: “The persons with chronic schizophrenia demonstrated much more RT, MT, and MU but less PV compared to healthy subjects when performing the reaching task” is very bad English and should be re-written. The whole manuscript also needs to be carefully edited by a native English speaker.

7. PLOS authors have the option to publish the peer review history of their article (what does this mean?). If published, this will include your full peer review and any attached files.

Reviewer #1: No

Reviewer #2: No

---

## [Author Response · Author response to Decision Letter 1]

26 May 2020

Date: May 17, 2020

To: Robin Baurès, Ph.D.

Manuscript ID number: PONE-D-19-26482

Dear Dr. Baurès:

Thank you very much for giving us the opportunity to re-revise our manuscript entitled “Effects of different types of sensory signals on reaching performance in persons with chronic schizophrenia” (ID numberPONE-D-19-26482). We also greatly appreciate the reviewers for their complimentary comments and suggestions. 

We have revised the manuscript as suggested by the reviewers except one point of the reviewer 2. We have provided reasons to support our claim. Please see the list of the revisions in the attached Detailed Response to Reviewers. We regret there were problems with the English. The paper has been carefully revised by a native English speaker to improve the readability.

Accordingly, we have uploaded a copy of highlighted version and a copy of clean version of the revised manuscript. 

We hope that you find our responses satisfactory and that the manuscript is now acceptable for publication.

Sincerely Yours,

Shu-Zon Lou, Ph. D

Assistant Professor,

Department of Occupational Therapy, Chung Shan Medical University

Detailed Response to Reviewers

Manuscript ID number: PONE-D-19-26482

Responses to Reviewer #1:

We appreciate that the reviewer’s comments. The followings are our point-by-point responses. Your comments are quoted in black font, our responses are in blue font and all the changes made in the revised manuscript are marked using the Microsoft Word's Track changes. The page and line numbers are in accordance with the highlighted version.

a few minor issues remain to be resolved:

abstract: final sentence is misleading. Authors did not test whether motor performance was linked to clinical status, except for CPZ and DOI. Please omit this information in the abstract

[Reply]: the final sentence”….. independent of clinical characteristics. “ has been changed to “… independent of duration of illness and antipsychotic dosage.” 

table 1 Levels of antipsychotics should be "dosage of antipsychotics" because authors do not report blood levels

[Reply]: corrected as suggested, Line 239 Table 1

results: again please delete the phrase on clinical characteristics (correlation with motor), but instead state that DOI and CPZ did not correlate (line 265 ff)

[Reply]: Line, 256, the sentence has been changed to “The results of Pearson’s correlations showed that reaching performance neither significantly correlated with duration of illness nor with antipsychotic dosage. “ 

discussion and throughout manuscript: please replace "schizophrenic patients" with "schizophrenia patients" or "patients with schizophrenia". Schizophrenic is considered to be a stigmatizing adjective

[Reply]: corrected as suggested

Detailed Response to Reviewers

Manuscript ID number: PONE-D-19-26482

Responses to Reviewer #2:

We appreciate that the reviewer’s comments. The followings are our point-by-point responses. Your comments are quoted in black font, our responses are in blue font and all the changes made in the revised manuscript are marked using the Microsoft Word's Track changes. The page and line numbers are in accordance with the highlighted version.

Reviewer #2: In this revised manuscript the authors have taken into consideration some of my previous comments but they did not address adequately several others and especially my criticism which is presented in comment number 3.

As I pointed out one cannot directly compare the time interval between successive movements in a self-paced sequence movement task which is the no signal condition with the externally triggered visual and auditory task in which the time of onset for each response is the reaction time. The authors just renamed RT in the no signal task to IRI. What they should have done is a separate analysis of RT in the visual and auditory conditions comparing patients and controls and a separate analysis for IRI in the self-paced movement task comparing the IRI between patients and controls. The discussion of these two measures should also be separate because different cognitive processes are engaged in self-paced sequential movements and different processes are engaged in externally driven RT processes. The authors do mention the relevant neural systems that are engaged in self-paced sequential movements that involve the prefrontal cortex and particularly area SMA. Indeed sequential movements are pre-programmed and executed in a unified manner. The authors observe a large difficulty of patients in initiating each component of the sequence of these sequential movements and this is very relevant to the difficulty of the patients in maintaining the sequence in working memory and initiating each movement in the sequence. This effect is different from the difficulty of the patients in initiating externally triggered movements to a target where the system involves the allocation of attention to the external stimulus and the programming of a response to this stimulus. A different fronto-parietal network is activated in these stimulus response tasks. In essence the authors observe differences between patients and controls in two different systems, one related to the programming of a sequence of movements and one to attending and responding to external stimuli. This is why I still think the authors should perform separate analyses for RT and IR and discuss the differences between patients and controls separately for RT and IRI.

[Reply]: You have raised an important point here and suggested that we should separate analysis of RT in the visual and auditory conditions comparing patients and controls and a separate analysis for IRI in the self-paced movement task comparing the IRI between patients and controls. However, we believe that combine the data to analyze would be more appropriate because combined a 2×2 two way ANOVA and one t-test to a 2×3 two way ANOVA allowed us to avoid the problem of multiple comparisons. In this study, the same experimental task (reaching out and return) was performed in three conditions. We don’t think it is inappropriate to directly compare reaching performance in the auditory, visual and self- paced target conditions.

In the literature, there were also some studies that compared self-initiated and externally triggered movements directly (Cunnington, et al., 2002, Jenkins, et. al., 2000, Gilbert, et al., 2009).

Cunnington, R., Windischberger, C., Deecke, L., & Moser, E. (2002). The preparation and execution of self-initiated and externally-triggered movement: a study of event-related fMRI. Neuroimage, 15(2), 373-385. doi:10.1006/nimg.2001.0976

Jenkins, I. H., Jahanshahi, M., Jueptner, M., Passingham, R. E., & Brooks, D. J. (2000). Self-initiated versus externally triggered movements: II. The effect of movement predictability on regional cerebral blood flow. Brain, 123(6), 1216-1228. doi:10.1093/brain/123.6.1216

Gilbert, S. J., Gollwitzer, P. M., Cohen, A. L., Burgess, P. W., & Oettingen, G. (2009). Separable brain systems supporting cued versus self-initiated realization of delayed intentions. J Exp Psychol Learn Mem Cogn, 35(4), 905-915. doi:10.1037/a0015535

Behavioral data. Left panel: Mean percentage of prospective memory (PM) targets detected in the self-initiated and cued conditions. Right panel: Mean reaction time (RT) in the ongoing tasks. Pale gray bars indicate RTs during the ongoing task at the beginning of the experiment, before prospective memory (PM) instructions were introduced. Black bars indicate reaction times after PM instructions were introduced. Error bars indicate standard error of the mean.

IN response to my comment on the effects of medication the authors report no correlation of medication to all measured variables. Could they provide the r coefficients in a table or a supplementary table?

[Reply]: Line 265, all the correlation coefficients has been shown in Table 2 

In response to my comment about the measurement of peak velocity and movement units there is still no clear answer. I commented in my review that the subjects perform two arm movements each time one reaching out to the target and one returning back to the home position and pressing the start button. The authors explained that the MT was the time from release of the home button to the pressing of it so MT corresponds to both arm movements. What about PV and MU? The authors say in their response that they get these measurements from the “zebris” system. That was not my question. My question was WHICH movement’s profile was used to measure these parameters? The reaching out movement or the returning movement or both?

[Reply]: Line 155, the sentence has been changed to “RT/IRI and MT of total movement (reaching out to touch and return) were measured by the Interactive Sound and Light Eye-Hand Coordination Training System, and the other velocity parameters (PV, PPV, and MU) of reaching out to touch were measured by Zebris.” for clarity.

The description of the results need rephrasing, for example the phrase: “The persons with chronic schizophrenia demonstrated much more RT, MT, and MU but less PV compared to healthy subjects when performing the reaching task” is very bad English and should be re-written. The whole manuscript also needs to be carefully edited by a native English speaker.

[Reply]: Line 246, the sentence has been changed to “ The persons with chronic schizophrenia showed significantly greater RT, MT, and MU but lower PV compared to healthy subjects when performing the reaching task.” 

I have sent my manuscript to https://www.servicescape.com/editors/acadconsult to edit before my first submission and I sent it to edit again before I re-submit the revised manuscript. The background of the editor is as follows, therefore, I supposed that she is a native English speaker and she is qualified to edit the manuscript.

AcadConsult (Gabriela Dye, PhD)

Credentials

• Ph.D. in Psychology from the University of Rhode Island

• M.A. in Psychology from the University of Rhode Island

• B.S. in Psychology from Old Dominion University

The editor re-read and edited the entire revised paper again. The paper has been carefully revised to improve the readability.

---

## [Decision Letter · Decision Letter 2]

8 Jun 2020

Effects of different types of sensory signals on reaching performance in persons with chronic schizophrenia

PONE-D-19-26482R2

Dear Dr. Lou,

We’re pleased to inform you that your manuscript has been judged scientifically suitable for publication and will be formally accepted for publication once it meets all outstanding technical requirements.

Kind regards,

Robin Baurès, Ph.D.

Academic Editor

PLOS ONE

Reviewers' comments:

Reviewer's Responses to Questions

**Comments to the Author**

1. If the authors have adequately addressed your comments raised in a previous round of review and you feel that this manuscript is now acceptable for publication, you may indicate that here to bypass the “Comments to the Author” section, enter your conflict of interest statement in the “Confidential to Editor” section, and submit your "Accept" recommendation.

Reviewer #2: All comments have been addressed

2. Is the manuscript technically sound, and do the data support the conclusions?

Reviewer #2: Yes

3. Has the statistical analysis been performed appropriately and rigorously? 

Reviewer #2: Yes

4. Have the authors made all data underlying the findings in their manuscript fully available?

Reviewer #2: Yes

5. Is the manuscript presented in an intelligible fashion and written in standard English?

Reviewer #2: Yes

6. Review Comments to the Author

Reviewer #2: I have no further comments that would require revision. All my comments were adequately addressed.

7. PLOS authors have the option to publish the peer review history of their article (what does this mean?). If published, this will include your full peer review and any attached files.

Reviewer #2: No

---

## [Editor Report · Acceptance letter]

12 Jun 2020

PONE-D-19-26482R2 

Effects of different types of sensory signals on reaching performance in persons with chronic schizophrenia 

Dear Dr. Lou:

I'm pleased to inform you that your manuscript has been deemed suitable for publication in PLOS ONE. Congratulations! Your manuscript is now with our production department. 

Kind regards, 

on behalf of

Dr. Robin Baurès 

Academic Editor

PLOS ONE